# Extending the Enterovirus Lead: Could a Related Picornavirus be Responsible for Diabetes in Humans?

**DOI:** 10.3390/microorganisms8091382

**Published:** 2020-09-10

**Authors:** William Klitz, Bo Niklasson

**Affiliations:** 1Department of Integrative Biology, University of California, Berkeley, CA 94720-3140, USA; 2Jordbro Primary Health Care Center, 137 64 Stockholm, Sweden; bo@boniklasson.se

**Keywords:** diabetes, picornavirus, Ljungan virus, T1D, T2D, animal models

## Abstract

We found an association between the abundance of rodents in the wild and onset of type 1 diabetes (T1D) in humans. A picornavirus named Ljungan virus (LV) was subsequently isolated from wild bank voles. Both picornavirus-like particles detected by electron microscopy and LV antigen visualized by immunohistochemistry was seen in islets of Langerhans in diabetic wild bank voles. LV antigen has also been found in islets of Langerhans in a patient with recent onset of T1D and in the commonly used Bio Breeding (BB) T1D rat model. We discuss the possibility of T1D and type 2 diabetes (T2D) as parts of a single disease entity. Antiviral compounds directed against picornavirus have been found to be an effective treatment of diabetes in BB rats. We propose using the same currently available antiviral compounds in clinical trials in humans. Antiviral treatment would have the potential to be both proof of concept for involvement of a picornavirus in diabetes pathogenesis and also present a first-generation therapy.

## 1. Introduction

Despite extensive research effort, the body of evidence supporting a relationship between viral infections and type 1 diabetes (T1D) remains largely circumstantial [1,2]. Support for the hypothesis that a virus can cause T1D in humans includes the fact that strains of enteroviruses and encephalomyocarditis virus, both belonging to the picornavirus family, have the ability to induce or accelerate diabetes in animal models [3,4]. The majority of reports on possible infectious etiology of T1D focus on picornavirus in general and enteroviruses in particular as playing a major role in development of T1D in humans. While meta-analysis of these reports demonstrates a significant excess of enterovirus infection associated with autoimmunity and T1D compared to controls, the proportion of T1D cases explained is modest, suggesting that a single picornavirus species may not be the typical culprit. In addition to picornavirus etiology, several other viral families have been proposed as an etiological agent for T1D, such as cytomegalovirus, parvovirus and retrovirus, all of which have been subsequently challenged.

In this review we present data suggesting that a new group of rodent-born picornaviruses, namely the Ljungan virus (including Ljungan related viruses), have the potential for explaining T1D in both animals and humans. We summarize studies and data generated during the past two decades both for and against the hypothesis that Ljungan virus (LV) or related virus causes diabetes in animals and in humans. We also introduce the idea of testing the hypothesis of picornavirus involvement in humans through antiviral clinical trials using compounds directed towards this viral family.

## 2. Ljungan Virus in the Picornavirus Family

In the early 1990s, a cluster of lethal cases of myocarditis occurred among elite orienteers in Sweden [5]. It was reasoned that exposure to an infectious agent might occur during pathfinding competitions in natural areas. A statistical study demonstrated that the incidence of lethal myocarditis in Sweden followed the three- to four-year population cycles characteristic of the small rodents in northern Sweden [6]. These fluctuations in small rodent population density led to the identification of rodents as vectors of Hantavirus infection (nephropathia epidemica) in Sweden in the 1930s and also for Korean hemorrhagic fever in the 1950s [7,8,9]. We used western blot analysis with serum-collected post mortem from lethal cases of myocarditis in parallel with serum from normal human controls to identify organs of potentially infected rodents trapped in the wild. Following a large number of virus isolation attempts, a novel picornavirus was isolated [10]. This first successful isolation used the protocol of intracerebral inoculation of one day old suckling mice. At signs of encephalitis, brain homogenate was passed into new suckling mice brain homogenate and was later passed into continuous cell line in tissue culture [10]. The fact that this group of viruses are very difficult to isolate in tissue culture and only replicates in a low to modest amount of viral particles without causing a clear cytopathogenic effect in tissue culture contributes to the diagnostic difficulties of this group of picornaviruses.

Initial sequence analysis of the non-coding region of the 5′end found LV to be related to Cardiovirus, while the sequence analysis of the coding region was related to Parechovirus (genus of human viral pathogens in the picornavirus family) [10]. Cardioviruses with rodents as their natural host and reservoir have a broad tissue tropism that causes clinical disease attributable to viral damage to the pancreatic islets, myocardial fibers causing myocarditis, and infection of CNS tissues causing meningoencephalitis and myelitis occurring in many animal species. Additionally, placental infection and fetal death also occurs in nonhuman primates and swine [11,12]. The finding that LV is related to both Cardiovirus with a track record of causing myocarditis, diabetes, neurodegenerative disease, and pregnancy-related diseases in many animal species and also related to the fact that Parechovirus being a human pathogen formed the hypothesis that LV may be a pathogen that causes the disease repertoire caused by Cardioviruses in humans.

The Ljungan virus replicates in a wide variety of different tissue culture cells, producing low or very low viral titers and a mild cytopathogenic effect often without visible cell lysis. Virus isolation in tissue culture requires multiple blind passages. A technique to detect viral proteins or viral RNA is needed to detect and follow viral replication during virus isolation attempts. The success rate of virus isolation attempts is low even from infected clinical specimens generated under experimental conditions [10,13,14]. Ljungan virus has been detected/isolated in several European countries, the USA, and Japan [10,15,16,17,18,19]. Genetic analyses of the virus strains showed that LVs can be divided into four genotypes [16,17,20]. LVs isolated in Sweden belong to genotypes 1 and 2, while North American strains are classified as genotypes 3 and 4 [16,20]. An LV isolate from wild bird feces collected in Japan has also been sequenced and suggested to be a novel genotype genetically distinct from known LVs [21]. The phylogenetic relationship between isolated Ljungan virus and other picornavirues is found on the Picornavirus webpage: https://www.picornaviridae.com/parechovirus/parechovirus.htm.

## 3. Ljungan Virus Associated with Diabetes in Animals

**Laboratory mice**. We have demonstrated that mice infected with LV developed a type 2 diabetes (T2D)-like disease [22], and have further shown that male mice developed T2D, including fat gain as adults, if infected during pregnancy with LV. Exposure to either LV or stress alone produced no diabetes. Only animals exposed to LV in utero and stress developed diabetes. Diabetes was more severe in mice whose mothers were infected earlier than in those whose mothers were infected later in pregnancy. This work suggests that a T2D-like disease in these rodents can be induced by a viral insult early in utero, and then set the stage for disease occurring later during adult life. It was also found that the combination of the antiviral drug Pleconaril and LV antisera could reduce indicators of the diabetic state by significantly reducing the blood glucose level (measured via a standard glucose tolerance test), lower abdominal fat weight, and by reducing serum insulin levels [23].

**Bank voles and other small wild rodents**. Schoenecker et al. found that bank voles in captivity developed both stereotypies and polydipsia and hypothesized that polydipsia may be a symptom of diabetes mellitus [24]. In a subsequent study it was shown that bank voles had diabetes mellitus, accompanied with auto-antibodies to glutamic acid carboxylase (GAD65), tyrosine phosphatase-like protein IA-2, and insulin-resembling characteristic markers of human T1D [25]. However, it was later found that glucose-intolerant diabetic bank voles had elevated levels of serum insulin when compared to animals classified as normal, thus, mimicking human T2D. Morphologic screening of pancreatic sections from bank voles across various ages did not reveal any clear signals of insulitis, which argues against classical human T1D pathogenesis. The condition observed in the glucose intolerant diabetic voles most closely resembles the human latent autoimmune diabetes in the adult (LADA) in humans, in which patients with a T2D phenotype exhibit islet autoantibodies [26,27]. It is possible that the diabetic condition in bank voles is first characterized more by a T2D phenotype, and later, in older animals, features of T1D may evolve. Symptoms and elements of both T1D and T2D are seen through a range of rodent models with β-cell destruction and fat deposition often seen in the same animal. The veterinary literature reveals such manifestations of both diabetes types in companion animals [28]. Here, we consider T1D and T2D as reflecting parts of a single disease entity with T1D and its active beta cell destruction as being the terminal phase of the disease.

The bank vole diabetes phenotype seems to depend to a large extent not only on the viral infection but also age and sex of the animal and the level and type of stress the animals are subjected to [24,25,29]. In contrast, the Cadiovirus mouse diabetes model offers another approach to explain the phenotype of diabetes, in which one clone of Cardiovirus can induce T2D in mice, while another clone causes T1D with the difference between the two viral strains due to only a few nucleotides [30,31].

The question as to whether LV infection causes or contributes to the development of diabetes in the bank vole has never been fully proved, as attempts to establish a virus-free bank vole colony has never been successful. However, the LV antigen was detected by immunohistochemistry in the islets of Langerhans of diabetic bank voles. In addition, picornavirus-like particles were visualized in the in islets of Langerhans in diabetic voles using thin-section transmission electron microscopy [32].

The observation that laboratory voles develop life threatening diabetes in the laboratory also raised the question as to whether this phenomenon occurs in the wild and if it may affect population dynamics. In screening for the disease in wild cyclic vole and lemming populations, we demonstrated that a high proportion of live-trapped bank voles, grey red-backed voles, field voles and lemmings at high collective peak density, and shortly before the population decline, suffered from diabetes or myocarditis in northern Sweden [33]. Extensive islet destruction was not seen in the wild voles at the time of capture, but did develop in the laboratory in a proportion of the animals in the following two months. Animals with symptoms of diabetes may be eliminated by predators, or perhaps such individuals are less likely to be trapped. We hypothesize that LV infection, either directly or indirectly, are involved in causing the regular, rapid population declines of these cyclic voles and lemmings recorded in previous long-term surveys.

**Bio Breeding rat type 1 diabetes model**. The Bio Breeding (BB) rat animal model of T1D originates from a Canadian colony of outbred Wistar rats spontaneously developing hyperglycemia that quickly progresses to fatal diabetic ketoacidosis unless treated with exogenous insulin. The BB rat is one of the most engaged animal models for the study of β-cell destruction and onset of T1D [34]. The fact that all BB rats develop diabetes within a narrow time frame provides an excellent opportunity to study the impact of treatment before, during, or after disease onset.

We observed the LV antigen in the islets of Langerhans in diabetic BB rats [35], thus suggesting LV infection intrinsic to the strain. This was followed by the finding that diabetes onset could be postponed using the antiviral compounds Pleconaril and Ribavirin [23,35]. We later extended our antiviral investigations using treatment with Pleconaril and an experimental antiviral kinase inhibitor APO-N039. The two compounds showed an additive effect when evaluated in tissue culture [36]. This finding was repeated in the BB rats with Pleconaril and APO-N039 administrated singly only showed marginal delay on diabetes onset, while the combination of both drugs protected all animals for the entire 6-week duration of treatment, suggesting possible virus elimination. Some animals remained non-diabetic several months post treatment. Some animals became diabetic when antiviral therapy was terminated but become non-diabetic again when antiviral therapy was reinstated. The BB rat has been perceived as an animal model in which autoimmunity is the main driving force in the pathogenesis [34]. However, the observation that the Pleconaril—APO-N039 combination could reverse already established clinical diabetes symptoms points to the possibility that diabetes can be treated after onset. The short duration between initiating antiviral treatment and clinical response and the clinical “on-off” effect observed when treatment was terminated and again reinstated suggests a direct effect of the antiviral compounds on a viral infection in the β-cells rather than a virus triggering an autoimmune effect. Assuming the scenario to be true, the immune system action of removing the insulin producing β-cells would be an appropriate response to an infected cell, which thereby induces diabetes instead of being part of an autoimmune process.

An intriguing observation is that the LV antigen detected in all diabetic BB rats disappeared in animals responding to antiviral therapy. Absence of LV antigen does not mean that an individual has cleared the infection, but can at least be interpreted as a significant decrease of the viral load in the islets of Langerhans. We have also recently presented an observation from human pancreas tissue originating from a fatal accident selected for organ donation of pancreas tissue [37]. Investigations prior to transplantation revealed that the patient had an undiagnosed recent onset of T1D. This in turn offered a unique opportunity to analyze islets of Langerhans from pancreas tissue in good condition at this critical stage of the disease. Formalin-fixed paraffin-embedded LV IHC microphotographs from this patient were stained in parallel with a pancreas specimen from a BB rat at the same stage of diabetes development, revealing striking similarities between the two species [38,39]. Presence of LV antigen using a polyclonal antibody and absence of PCR confirmation may depend on the low specificity of the PCR used.

The sensitivity and specificity of any diagnostic PCR assay are dependent on the sequence information available from sequenced viruses representing the genetic variation for the viral disease to be diagnosed. When a new group of viruses is to be defined, consensus primers are used for the diagnosis of clinical specimens. The specificity of any PCR usually improves gradually over time, when an increasing number of isolates representing the full genome variation are being characterized, sequenced, and used for primer design [40]

We have recently extended our experiments in BB rats using Pleconaril, Ribavirin, and Efavirenz, three antiviral compounds selected based on potential effects on an LV-related picornavirus and are all compounds that can be used in humans [37].

Pleconaril binds to a hydrophobic pocket in the viral protein 1, the major protein comprising the capsid of picornaviruses. This binding renders the viral capsid rigid, compressed and prevents cell entry, un-coating of RNA and also assembly of new viral particles. As a result, the virus is stopped from this critical phase of the infectious process [41]. LV belongs to the Parechovirus genus in the picornavirus family and several members of the Parechovirus including Ljunganvirus lack a hydrophobic pocket binding structure on the surface of the VP1 protein [42,43]. Despite this finding, Pleconaril does show antiviral activity to LV in tissue culture [36]. Ribavirin, a nucleotide analogue of guanosine, has broad-spectrum direct antiviral effect on members of the picornavirus family. Ribavirin was also selected based on antiviral activity to LV in tissue culture and apparently acting through distinct aspects of the viral life cycle [23,44]. A dose-dependent toxicity resulting in hemolytic anemia is a limiting side effect of this drug. Efavirenz, a non-nucleoside reverse transcriptase inhibitor, is widely used against HIV [45]. The antiviral effect on picornavirus is unexpected, since the picornavirus does not utilize reverse transcriptase for replication, suggesting an additional antiviral mechanism not yet identified. The combination of Pleconaril and Efavirenz was superior as compared to Pleconaril alone. Pleconaril-Efavirenz and Ribavirin added additional delays to diabetes onset. The general advantage of combination therapy over monotherapy is supported by both theoretical models of virus dynamics and treatment experience [46].

## 4. Ljungan Virus Associated with Diabetes in Humans?

In the 1990s, we proposed that type 1 diabetes in humans may be due to infectious agents carried by rodents, based on the observations of an association between rodent density and disease incidence [6]. Long-term records (>10 years) on small rodent abundance are scarce, but in Sweden such data are available from the area with cyclic rodent populations [47]. We studied T1D in an area of northern Sweden with such cyclic rodent populations. Diabetic patients were identified through registries of patients with diabetes in regional hospitals and included all age groups. A total of 318 cases of IDDM were recorded in 1971 to 1991. We found that changes of the T1D numbers from one year to the next varied with changes in vole abundance two years previously, suggested by high positive and significant correlations [6]. These observations initiated attempts to isolate new viruses from small rodents, the long-term goal of which was to find new etiologic agents causing diseases in humans.

The ability to identify and then associate a virus with human disease is influenced in part by its biological characteristics. Difficult to cultivate viruses, such as LV, decrease the likelihood of linking the agent to disease. A zoonotic agent has little selective pressure for efficient spread beyond the newly infected host individual. This may be why many zoonotic viruses are identified through more challenging procedures of first being isolated in a natural reservoir. Previously reported observations of LV viral antigen in human tissue have never been definitely confirmed by virus isolation or detection of viral RNA using PCR. Stool samples from children participating in the Norwegian Environmental Trigger of Type 1 Diabetes Cohort study found all samples LV negative [48]. A Finnish Diabetes Prediction and Prevention study found no association between LV seropositivity and the β-cell damaging process [49]. In contrast, a Swedish study reported the presence of LV antibody representing prior exposure to LV to be associated with T1D and also that LV antibodies in young children correlated with both insulin autoantibodies and HLA-DQ8, suggesting a possible role in T1D pathogenesis [50,51].

Based on our experience with the BB rat model of T1D treatment, we suggest human trials for T1D utilizing antiviral drugs currently available for human use. Dosing and possible safety considerations could be carefully followed along with control of the diabetic state itself. A placebo-controlled trial of antivirals to control T1D is currently taking place in Norway. We eagerly await the results of that effort [52,53,54].

## 5. Concluding Remarks

Nearly a century has passed since T1D was first identified as a disease of the pancreas and successfully treated with insulin. Despite many decades of intensive biomedical research, no underlying cause/causes has been brought forward to explain the disease beyond a characteristic adaptive immune response directed against insulin producing cells involving B and T cells, which was later attributed to autoimmunity.

This review of research on the underlying cause of diabetes mellitus as due to a picornavirus—the picornavirus conjecture—addressing work ranging across wild and laboratory animal models and then to humans themselves. Beginning with the discovery of a new picornavirus in native European rodents, we cover possible factors influencing viral transmission, infectious models, the importance of stress, and the apparent biological unity of diabetes mellitus types. Finally, given the infectious etiology, we offer a new treatment approach in humans now ready for trial.

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
