# Peer review of "Extending the Enterovirus Lead: Could a Related Picornavirus be Responsible for Diabetes in Humans?"

_microorganisms, 2020, doi:10.3390/microorganisms8091382_

Round 1

Reviewer 1 Report

The review represents a relevant contribution to the viral origin of diabetes, especially (but not only) insulin-deficient diabetes. Authors are well qualified in the field and have adequately summarized and highlighted significant points of their own research on Ljungan viruses and related agents.

Significance of the MS would increase if authors would insert additional information (short text and Figures) regarding:

1) biological properties of Ljungan viruses that distinguish them from other Picornaviruses, mainly cardioviruses, enteroviruses, parechoviruses. Species-specificity and tissue tropism should also be treated;

2) phylogenesis of different LV genotypes;

3) LV types detected in the BB rats (a common model of virus-associated diabetes);

4) list and properties of drugs potentially effective against LV infection;

5) possible effect of LV vaccine in animals.

Literature is adequate, English language needs only minor revision.

This review may become a reference and a guide to those studying virus-induced diabetes in humans.

Author Response

#1 Reviewer requests additional information on LV properties and specificity (points 1-3).

We have added several points pertinent to the request:

Section 2 lines 47-52, 88-97, 205-218.

#2 An LV phylogeny from the Picornavirus website is now given.

Section 2 lines 73-85.

#3 Request for LV sequences detected in BB rat experiments. We have no sequence information from these experiments. Our evidence of LV-like virus presence is dependent on positive reactions to LV antigens.

#4 Drugs effective against LV infection. Antivirals are discussed in the last paragraph of Section 3.

#5 While vaccination is in the back of the mind of every viral epidemiologist, there is no insight as to an LV vaccine or its possible use at this time.

Reviewer 2 Report

This review discusses the interesting concept of picornavirus and the development of diabetes.

I think that is a great story and should be published. I particularly enjoyed the epidemiology in section 4.

I have, however, one or two concerns (some of these are partially adressed in the paper).

  • Implicit in the title is type 1 diabetes yet there are mentions of type 2 diabetes and LADA
  • In section 3 (3.1) there is discussion about the development of a T2DM-like disease in laboratory mice
  • In section 3 (3.2) there is discussion about the development of elevated anti-GAD and anti-IA2 in bank voles etc, which is characteristic of type 1 diabetes.  Yet these animals had high serum insulin levels mimicking T2DM.
  • The anti-viral agents stop β-cell destruction by the virus rather than stopping the virus triggering an autoimmune process.
  • There is mention of four genotypes in the final part of section 2 but no further mention of what the difference is in the four genotypes (other than geography).
  • Do anti-viral medications have any other mechanism(s) of action to explain their role in protecting the BB rats from diabetes?
  • In section 3.3 para 3, there is mention of the PCR failing to demonstrate picornavirus. The explanation would have increased strength if it were referenced.
  • I would significantly shorten paragraph 2 in the Concluding remarks and take out “appearance of concentration camp survivors to fleshed-out….” and have a couple of sentences about the possibility of picornavirus having different effects depending on when it infects and the stress the animal is under.

Author Response

We agree with reviewer 2 that observations in wild rodents and laboratory mouse experiments support the ongoing debate and question whether T1D and T2D are two different diseases, or if T1D and T2D might be variable manifestations of a single viral infection, expressed at different times and in different tissues.  Our data give support for the latter alternative in this debate. Our data also support the idea that T1D in BB rats is not caused by an autoimmune reaction, but instead is a result of cell destruction directly due to the virus.  Our attempt has been to describe this hypothesis in a balanced way. (see Section 3 paragraph 2 along with new reference (lines 112-116).

We have inserted the following lines (and reference) at the end of Para 3.3.  “Symptoms and elements of both T1D and T2D are seen through a range of rodent models with beta cell destruction and fat deposition often seen in the same animal. The veterinary literature reveals manifestations of both diabetes types in companion animals [Gilor et al, J Vet Intern Med. 2016 30(4):927-940.]. Here we consider T1D and T2D as reflecting parts of a single disease entity with T1D and its active beta cell destruction as being the terminal phase of the disease.”

In section 3.3 regarding the fact that we fail to confirm results by PCR we have edited and expanded the text and also added a reference as suggested.

We have significantly shortened section 5 (Concluding remarks) as suggested.

We have added a short paragraph to the end of Section 4, making explicit the idea of beginning human trials with antivirals for the control of T1D. Finally, we also point out a current antiviral drug trial taking place in Norway (end Section 4).